# Transparent flexible thermoelectric material based on non-toxic earth-abundant p-type copper iodide thin film

C. Yang[1], D. Souchay[2], M. Kneiß[1], M. Bogner[3,4], H.M. Wei[1], M. Lorenz[1], O. Oeckler[2], G. Benstetter[3], Y.Q. Fu[4,5] & M. Grundmann[1]

Thermoelectric devices that are flexible and optically transparent hold unique promise for future electronics. However, development of invisible thermoelectric elements is hindered by the lack of p-type transparent thermoelectric materials. Here we present the superior room-temperature thermoelectric performance of p-type transparent copper iodide (CuI) thin films. Large Seebeck coefficients and power factors of the obtained CuI thin films are analysed based on a single-band model. The low-thermal conductivity of the CuI films is attributed to a combined effect of the heavy element iodine and strong phonon scattering. Accordingly, we achieve a large thermoelectric figure of merit of $ZT = 0.21$ at 300 K for the CuI films, which is three orders of magnitude higher compared with state-of-the-art p-type transparent materials. A transparent and flexible CuI-based thermoelectric element is demonstrated. Our findings open a path for multifunctional technologies combing transparent electronics, flexible electronics and thermoelectricity.

[1] Felix-Bloch-Institut für Festkörperphysik, Universität Leipzig, Linnéstr. 5, Leipzig 04103, Germany. [2] Institut für Mineralogie, Kristallographie und Materialwissenschaft, Universität Leipzig, Scharnhorststr. 20, Leipzig 04275, Germany. [3] Deggendorf Institute of Technology, Edlmairstr. 6 + 8, Deggendorf 94469, Germany. [4] Faculty of Engineering and Environment, Northumbria University, Newcastle upon Tyne NE1 8ST, UK. [5] Institute of Fundamental and Frontier Sciences, University of Electronic Science and Technology of China, Chengdu 610054, China. Correspondence and requests for materials should be addressed to C.Y. (email: yangchangyc@gmail.com).

Thermoelectric energy recovery in room temperature range has attracted significant interest because of the usages of numerous heat sources near room temperature such as consumer electronics, solar cells, home heating and wearable devices on human bodies. Generally, heavily doped semiconductors with an energy bandgap value $E_g$ close to $10k_BT_O$ can be used as good thermoelectric materials, where $k_B$ is the Boltzmann constant and $T_O$ the operating temperature[1]. As a matter of fact, most established thermoelectric materials tend to be optically opaque due to their small bandgap ($E_g \ll 2\,eV$), such as the $(Bi,Sb)_2Te_3$ family[2,3]. Several classes of non-transparent thermoelectric materials with attractive thermoelectric properties for renewable power generation applications were recently reported, including tellurides[4], half-Heuslers[5] and silicides[6]. Up to date, few optically transparent ($E_g > 3\,eV$) thermoelectric devices are known to exist, whereas realization of such invisible devices could open new fields in a range of novel applications such as smart windows (or screens) with energy harvesting, cooling and thermal sensing functionalities. Another potential application of transparent thermoelectric elements is the fast on-chip cooling and power recovery[7,8] for optoelectronic devices including solar cells, infrared photodetectors, as well as fully transparent electronic devices.

Good thermoelectric properties are expected for a large dimensionless figure of merit ($ZT$).

$$ZT = S^2\sigma T\kappa^{-1} \qquad (1)$$

The value of $ZT$ is calculated from the Seebeck coefficient $S$, electrical conductivity $\sigma$, temperature $T$ and thermal conductivity $\kappa$. Accordingly, a transparent thermoelectric material is primarily a transparent conductor (TC) with a high value of $S$ and a low value of $\kappa$. Since the conventional thermoelectric modules consist of coupled n- and p-type thermoelectric legs, both n- and p-type transparent thermoelectric materials are required. However, due to the lack of highly conductive p-type TCs, recent research has focused on thermoelectric properties of n-type TCs including heavily doped ZnO (ref. 9), $In_2O_3$ (ref. 10) and $SrTiO_3$ (ref. 11). Among these n-type TCs, Sn-doped $In_2O_3$ (ITO) shows the highest value of $ZT \sim 0.14$ at room temperature[10]. On the other hand, p-type TCs usually exhibit poor electrical conductivities $\sigma$, which are several orders of magnitude smaller than those of n-type TCs, leading to poor thermoelectric performance at room temperature, for example, $ZT \sim 0.001$ for $CuAlO_2$ (ref. 12) and $ZT \sim 0.002$ for $CuCrO_2$ (ref. 13). The p-type misfit-layered oxide $Ca_3Co_4O_9$ was reported to possess thermoelectric performance ($ZT \sim 0.07$ at room temperature for single crystal) comparable to those of the n-type TCs, but it is not truly transparent due to its small bandgap ($E_g \sim 2.1\,eV$)[14]. Consequently, the lack of p-type transparent thermoelectric materials is the main obstacle in realization of fully transparent thermoelectric devices.

In this study, we focus on the room-temperature thermoelectric performances of copper iodide (CuI) thin films, since CuI in its ground-state phase was recently reported as a high performance p-type TC material[15-17]. CuI is an environment-friendly material composed of non-toxic and naturally abundant elements. Present in sea water, iodine is the heaviest essential element used widely in daily life for biological functions. Interestingly, the presence of heavy elements like iodine leads to a low thermal conductivity, which is necessary for a good thermoelectric performance. CuI undergoes multiple polymorphic phase transitions during heating, that is, $\gamma$ (zincblende structure, space group $F\bar{4}3m$) $\rightarrow$ $\beta$ (wurtzite structure, $P6_3mc$) $\rightarrow$ $\alpha$ (rock salt structure, $Fm\bar{3}m$) at 643 K and 673 K, respectively[18]. In the $\alpha$- or $\beta$-phase, it acts as a superionic conductor which is regarded as a phonon–liquid material with a low-thermal conductivity[19]. Whereas in its $\gamma$-phase at room temperature

(or below 643 K), it is a wide direct bandgap ($E_g = 3.1\,eV$) semiconductor with p-type conductivity. The copper vacancy, which has a shallow acceptor level, is primarily responsible for p-type conductivity of $\gamma$-CuI (ref. 15). Very recently a record high room-temperature hole conductivity $\sigma > 280\,S\,cm^{-1}$ has been achieved for CuI thin film[17]. At 300 K, a large Seebeck coefficient of $\gamma$-CuI ($S = 237\,\mu V\,K^{-1}$ for a hole concentration $p$ of $10^{20}\,cm^{-3}$) was theoretically predicted based on the Boltzmann transport theory combining with first-principles calculations[20]. Consequently, a high thermoelectric performance in the wide bandgap $\gamma$-CuI can be expected owing to its high values of $\sigma$ (with respect to the large bandgap) and $S$, as well as the predicable low value of $\kappa$. For the purpose of exploring truly transparent p-type thermoelectric materials, the thermoelectric properties of $\gamma$-CuI thin films have been investigated in this work. We report the superior thermoelectric performance of transparent p-type $\gamma$-CuI thin films in room temperature range. The thermoelectric transport properties of $\gamma$-CuI are analysed based on the single-band model. Furthermore, a prototype of CuI-based transparent and flexible thermoelectric module is demonstrated.

## Results

**Superior thermoelectric performance of CuI thin films.** $\gamma$-CuI is a native p-type conductor due to the existence of copper vacancies with a high hole mobility ($\mu > 40\,cm^2\,V^{-1}\,s^{-1}$ in bulk material) owing to the fairly small effective mass of 0.30 $m_0$ for light holes[15,21]. The heavy-hole band of $\gamma$-CuI does not explicitly reduce the hole mobility $\mu$ because it degenerates with the light-hole band under a strain-free condition. The single type of carrier (hole) in $\gamma$-CuI is favourable for a large Seebeck coefficient $S$ (ref. 22). The obtained $\gamma$-CuI samples in this study (see Supplementary Note 1 and Supplementary Fig. 1) are polycrystalline thin films deposited on glass and flexible polyethylene terephthalate (PET) substrates, which is consistent with our previous reports[16,17]. The hole concentration of $\gamma$-CuI was controlled by varying the iodine partial pressure in the sputter chamber during the deposition[17]. Figure 1a shows the $S$ values of $\gamma$-CuI on glass as a function of hole concentration $p$ at 300 K. The plot of experimental values of $S$ versus $p$ from this work (red squares in Fig. 1a) follows precisely the reported theoretical trend obtained based on the Boltzmann theory[20] (dashed line in Fig. 1a). These theoretical $S$ values are relatively over-estimated due to the under-estimated bandgap (theoretical $E_g = 2.23\,eV$) in the first-principles calculations using local or semi-local approximations[20]. The difference between our experimental data and theoretical data is about $75\,\mu V\,K^{-1}$. For the experimental data reported in reference[23], the $S$ value of an as-prepared $\gamma$-CuI thin film is very close to our data, but becomes smaller than our $S$ values after post-annealing. Higher annealing temperature leads to a lower value of $p$ and even larger deviations in value of $S$. Such deviations might be related to the formation of defects (that is, iodine vacancies and electrically neutral defect complexes) during the post-annealing[23].

The Seebeck coefficient $S$ of a p-type semiconductor can be obtained using the Pisarenko formula[1]:

$$S = \frac{k_B}{e}\left(A + \ln\frac{2(2\pi m^* k_B T)^{3/2}}{ph^3}\right) \qquad (2)$$

where $A$ is the scattering parameter, $p$ is the hole concentration, $m^*$ is the effective mass of the carrier, $e$ is the elementary charge and $h$ is the Planck constant[1]. For $\gamma$-CuI, the theoretical (directionally averaged) effective masses of electrons and light holes are 0.30 $m_0$ and those of the heavy holes are 2.14 $m_0$ (ref. 21). Thus, the large $S$ value of the $\gamma$-CuI is attributed to a large effective mass $m^*$ which is dominated by the heavy-hole band. According to the theoretical $S$ data (dashed line in Fig. 1a),

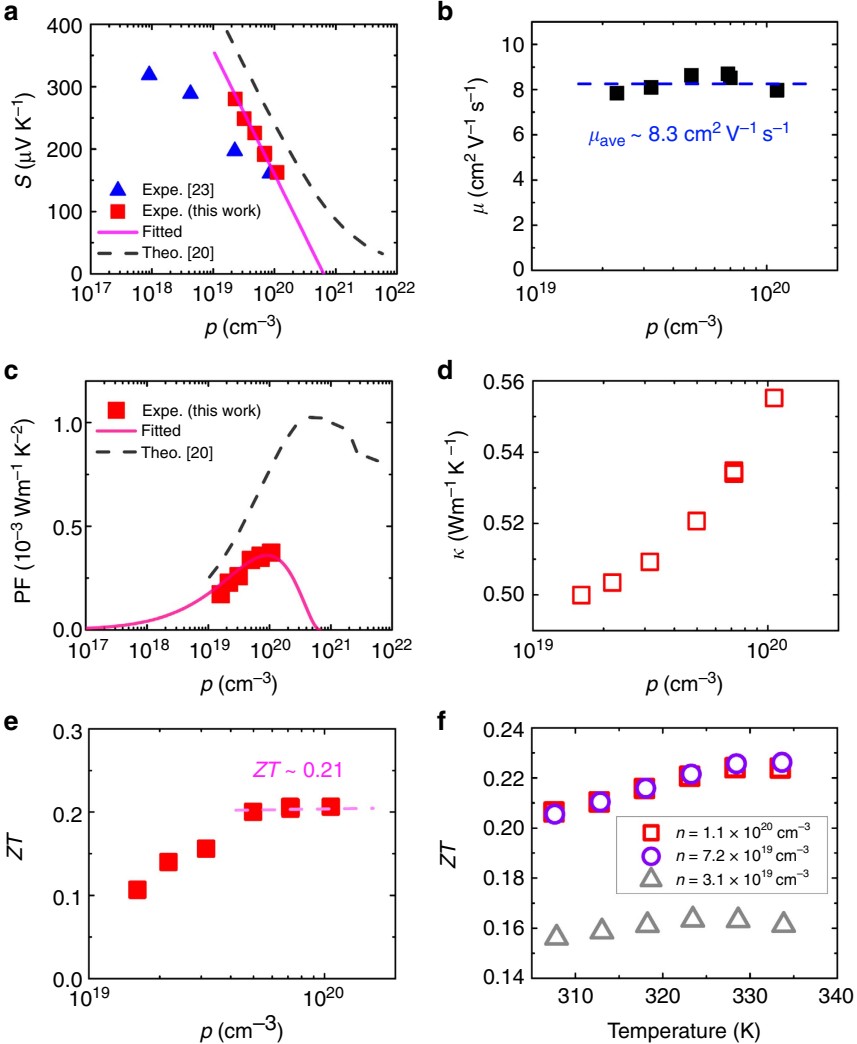

**Figure 1 | Thermoelectric properties of the p-type CuI thin films.** (**a**) Seebeck coefficients $S$, (**b**) hole mobility $\mu$, (**c**) power factor PF, (**d**) thermal conductivity $\kappa$ and (**e**) figure-of-merit $ZT$ of $\gamma$-CuI thin films (thickness of 300 nm) deposited on glass as a function of hole concentration $p$ at 300 K. (**f**) Figure-of-merit $ZT$ of $\gamma$-CuI thin films deposited on glass near room temperature. In **a** the solid line is a fit with equation (4). In **b** and **e** the dashed line is a guide to eye indicating the averaged hole mobility $\mu_{ave}$ or the saturation of $ZT$. In (**c**) the solid line is derived from the calculation using equation (6) with a single-band model, and the dashed line presents the theoretical data taking into account the non-parabolicity of the bands.

the flattening phenomenon at around $100\,\mu\text{V K}^{-1}$ for hole concentrations greater than $5\times10^{20}\,\text{cm}^{-3}$ is related to the increased effective mass $m^{*}$ of heavily doped $\gamma$-CuI. In such a high $p$ region, the large value of hole density would move the Fermi level deeper into the valance band, resulting in a higher effective mass $m^{*}$. However, due to the limited hole density of $P<2\times10^{20}\,\text{cm}^{-3}$ in this work, the value of hole mobility $\mu$ (Fig. 1b) does not vary much within such a hole density range, indicating a relatively constant effective mass $m^{*}$. Hence, the single-band model is adopted here for the Boltzmann approximation and a parabolic valance band[24], thus resulting in a linear relationship of $S$ versus log $p$. Accordingly, equation (2) can be rewritten into:

$$S = \frac{k_{B}}{e}\left(A + \ln\frac{2(2\pi m^{*}k_{B}T)^{3/2}}{h^{3}}\right) - \frac{k_{B}}{e}\ln p \quad (3)$$

or can be simplified into:

$$S = C - (86.17\,\mu\text{V K}^{-1})\ln(p\,\text{cm}^{-3}) \quad (4)$$

where the plot of $S$ versus $\ln p$ shows a straight line with a slope of $-k_{B}/e$ or $-86.17\,\mu\text{V K}^{-1}$. Thereby a Pisarenko plot (solid line

in Fig. 1a) can be derived from equation (4) with a fixed slope of $-86.17\,\mu\text{V K}^{-1}$, which fits well with the experimental data in this work. The fitting parameter $C$ is equal to $4128\pm3\,\mu\text{V K}^{-1}$. Accordingly, the value of scattering parameter $A$ can be determined to be 2.05, indicating ionized-impurity scattering in the obtained $\gamma$-CuI thin films[15].

For p-type semiconductors with negligible electron conduction, the conductivity value $\sigma$ can be obtained from:

$$\sigma = e\mu p \quad (5)$$

Therefore, the value of thermoelectric power factor $\text{PF}=S^{2}\sigma$ can be calculated by combining equations (4) and (5) as:

$$\text{PF} = e\mu p\left[4128\,\mu\text{V K}^{-1} - (86.17\,\mu\text{V K}^{-1})\ln(p\,\text{cm}^{-3})\right]^{2} \quad (6)$$

The value of $\mu$ is considered here as a constant with an average value of $8.3\,\text{cm}^{2}\,\text{V}^{-1}\,\text{s}^{-1}$, as shown in Fig. 1b. With this approximation, the PF can be calculated using equation (6) as a function of $p$. Figure 1c shows the calculated room-temperature PF values of the $\gamma$-CuI using the single-band model (solid line), which are consistent with the experimental results. The estimated maximum PF value is $3.59\times10^{-4}\,\text{W m}^{-1}\,\text{K}^{-2}$ at which

$p = 1 \times 10^{20}\,\mathrm{cm^{-3}}$ and $S = 172\,\mu\mathrm{V\,K^{-1}}$. The highest experimental value of PF in this work is $3.75 \times 10^{-4}\,\mathrm{W\,m^{-1}\,K^{-2}}$ at 300 K, which is comparable with those of the n-type TCs such as Al-doped ZnO ($3.9 \times 10^{-4}\,\mathrm{W\,m^{-1}\,K^{-2}}$) (ref. 9).

Because of the relatively low-hole concentrations of the films in this study, we could not further evaluate the thermoelectric properties of heavily (intrinsic) doped $\gamma$-CuI. The dashed line in Fig. 1c presents the theoretical values of PF at room temperature taking into account the non-parabolicity of the bands[20] for heavily doped $\gamma$-CuI, indicating that even larger PF values are achievable for highly degenerate $\gamma$-CuI (with hole concentration $p \gg 1 \times 10^{20}\,\mathrm{cm^{-3}}$). We note that the degeneracy of the heavy- and light-hole bands of $\gamma$-CuI is beneficial for a good thermoelectric performance because it simultaneously leads to a high Seebeck coefficient $S$ and a high electrical conductivity $\sigma$.

To determine the $ZT$ value, knowledge of the thermal conductivity $\kappa$ of the $\gamma$-CuI thin film is required. The lower the value of $\kappa$, the higher the $ZT$ will be. However, the $\kappa$ value of $\gamma$-CuI has never been reported to the best of our knowledge. In general, the thermal conductivity $\kappa$ of a semiconductor originates from two sources: (1) $\kappa_e$, which is linked to heat transport from electron and hole current and (2) $\kappa_{ph}$, which is linked with phonon scattering. According to the Wiedemann–Franz law[25], $\kappa$ is given by:

$$\kappa = \kappa_e + \kappa_{ph} = \sigma L T + \kappa_{ph}, \qquad (7)$$

where $L$ is the Lorenz factor, $2.4 \times 10^{-8}\,\mathrm{W\,\Omega\,K^{-2}}$ for ideal metals and $1.5 \times 10^{-8}\,\mathrm{W\,\Omega\,K^{-2}}$ for ideal non-degenerate semiconductors[25]. Here the non-degenerate $L$ is adopted, and thus the calculated value of $\kappa_e$ for $\gamma$-CuI ranges from 0.02 to $0.06\,\mathrm{W\,m^{-1}\,K^{-1}}$ depending on the hole concentration $p$. On the other hand, the value of $\kappa_{ph}$ is independent of electrical

properties. By using a developed 3ω method in this study[26], the total thermal conductivity $\kappa$ of the CuI sample with PF of $3.75 \times 10^{-4}\,\mathrm{W\,m^{-1}\,K^{-2}}$ has been determined, as low as $0.55\,\mathrm{W\,m^{-1}\,K^{-1}}$ near room temperature. This $\kappa$ value is even lower than that of glass[27] or phonon–liquid material $Cu_2Se$ (ref. 19) ($\kappa \sim 1.0\,\mathrm{W\,m^{-1}\,K^{-1}}$ at room temperature). Thereby, we can estimate a $\kappa_{ph}$ value around $0.5\,\mathrm{W\,m^{-1}\,K^{-1}}$. Such a low $\kappa_{ph}$ value is attributed to the heavy element iodine, as well as the strong phonon scattering at grain boundaries due to the polycrystalline nature of the $\gamma$-CuI thin films. The total thermal conductivity $\kappa$ of the $\gamma$-CuI thin film with different hole concentrations $p$ can be estimated by equation (7), as shown in Fig. 1d, where the $\kappa$ value slightly increases with increasing $p$.

Subsequently, the thermoelectric $ZT$ values of the $\gamma$-CuI thin films on glass are calculated using equation (1). Figure 1e shows the carrier density dependence of $ZT$ at room temperature. The values of $ZT$ increase with increasing hole density $p$, and then reach a saturation point at $P > 5 \times 10^{19}\,\mathrm{cm^{-3}}$. The maximum $ZT$ at 300 K is around 0.21, which is comparable with that of p-type PbTe ($ZT \sim 0.3$ in bulk[28]). The thermoelectric performance of the CuI thin films could be further enhanced at slightly elevated temperatures, as shown in Fig. 1f. The values of $\sigma$ and $S$ for these films can be found in Supplementary Fig. 2. The room-temperature $\kappa$ value was used to calculate the $ZT$ values at such slightly elevated temperatures. The value of $ZT$ is estimated up to 0.23 at around 330 K. Considering the diffusion of iodine (and other defects) in $\gamma$-CuI thin films during heat treatment, as described in ref. 23, higher heating temperatures over 350 K frequently lead to degraded electrical conductivity of $\gamma$-CuI. Hence, a capping (protective) layer is required to improve the thermal stability of $\gamma$-CuI thin film at further elevated temperatures. For example, as shown in Supplementary Note 2

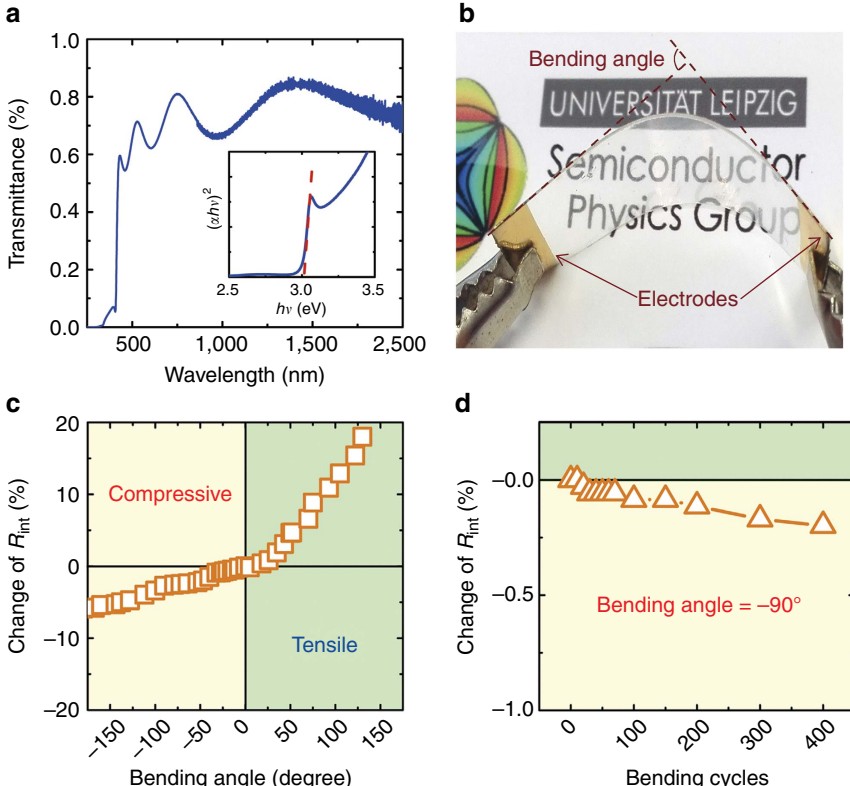

**Figure 2 | Transparency and flexibility of the CuI thin films.** (**a**) Transmittance spectra of CuI thin film (thickness of 300 nm) deposited on glass. Inset is the $(\alpha h\nu)^2$ versus $h\nu$ plot of CuI thin film. (**b**) The actual photo and (**c**) internal resistance $R_{int}$ stability of a CuI thin film sample deposited on PET with different bending angles. (**d**) The reliability test by repeated bending cycles.

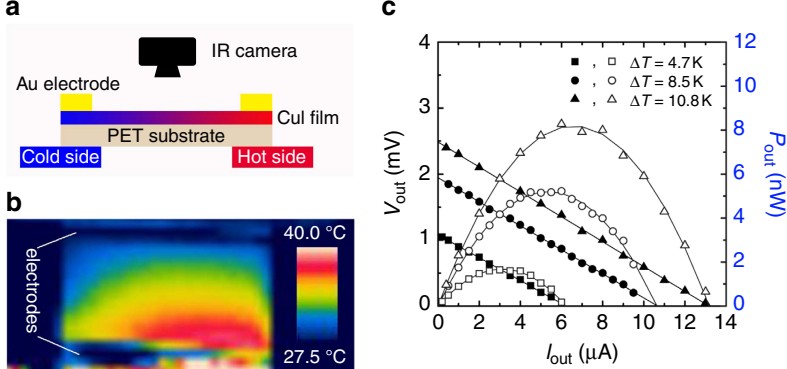

**Figure 3 | A CuI-based single-leg thermoelectric module.** (**a**) Schematic illustration for the power output measurement of the CuI/PET single-leg thermoelectric device. (**b**) The example infrared image taken during one of the measurements. The electrical contacts appear cold in the image because of their different emissivity compared with CuI. (**c**) Output voltage $V_{out}$ and output power $P_{out}$ of a CuI-based thermoelectric leg as a function of output current $I_{out}$ for three temperature differences.

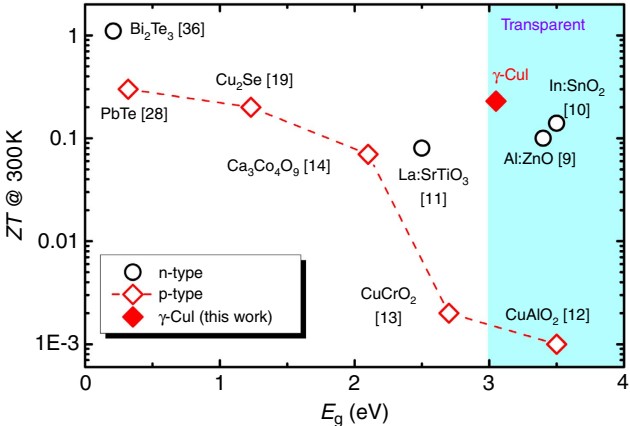

**Figure 4 | Figure of merit versus energy bandgap.** Comparing the thermoelectric $ZT$ of typical n- and p-type thermoelectric materials and γ-CuI thin films (this work) at room temperature. Dashed line is guide to the eyes. Detailed data see Supplementary Table 1.

and Supplementary Fig. 3, the degradation of the conductivity of γ-CuI thin film during air-annealing could be significantly suppressed by capping with a $CuO_x$ layer. It shows a three times reduced effect of increasing resistance and thus proves that further technological work targeted towards capping can extend the range of useful operation temperatures. Detailed study on the thermal stability of CuI are subject to further investigations by design and development of suitable capping layers. Although high-temperature thermoelectric performance of CuI is beyond the scope of this study, we note that molten CuI has been found to exhibit a large value of $ZT$ over 0.1 due to the large Seebeck coefficient 620–890 μV K$^{-1}$ and small thermal conductivity 0.57 W m$^{-1}$ K$^{-1}$ at high temperature (900–1150 K)[29].

**The transparency and flexibility of CuI thin films.** The optical transparency of the γ-CuI sample can be confirmed by the optical transmittance spectra as shown in Fig. 2a, where the transmittance, exhibiting interference effects due to film thickness, is ~60–85% over the visible and near infrared regions (wavelength of 410–2,000 nm). We also estimate the bandgap $E_g$ of the obtained γ-CuI sample using Tauc's equation[30]:

$$\alpha h v = B(h v - E_g)^m \tag{8}$$

where α is the optical absorption coefficient, $hv$ is the photon energy, B is a constant, m is 1/2 for a direct band transition and 2 for an indirect band transition[30]. The inset in Fig. 2a presents a linear relationship between $(\alpha h v)^2$ and $hv$, indicating a direct energy bandgap of the γ-CuI thin film. The value of $E_g$ is determined to be about 3.02 eV, which is consistent with that of bulk γ-CuI (ref. 15). Such a wide direct bandgap $E_g > 3.0$ eV allows for the full transparency of γ-CuI in the visible spectral range.

The mechanical flexibility of γ-CuI film deposited on PET has been investigated, and the results are shown in Fig. 2b–d. We found that compressive stress leads to an improved electrical conductivity, whereas tensile stress results in a degraded electrical conductivity. This phenomenon can be explained by the enhanced (or worsened) compactness of the grains in the polycrystalline γ-CuI thin film under a compressive (or tensile) strain. The change of the internal resistance $R_{int}$ is <3% with a compressive bending angle up to 90°, and <0.2% after repeatedly bent up to 400 cycles. These results indicate that the CuI-based thermoelectric modules can properly work on curved thermal energy sources such as hot-water tubes, clothes and human bodies.

**A transparent single-leg thermoelectric module.** A prototype of the transparent thermoelectric module has been fabricated as a demonstrator, which consists of a single thermoelectric leg of γ-CuI thin film on PET substrate, as illustrated in Fig. 3a,b. We note that practical thermoelectric devices are constructed by coupling the n- and p-type thermoelectric legs. Here the single-leg setup is only used to evaluate the output characteristics of CuI-based thermoelectric modules. The output characteristics of such a demonstrator are shown in Fig. 3c for three temperature differences (ΔT = 4.7, 8.5 and 10.8 K), and the simulated curves fit the experimental data well. This single thermoelectric leg generates a maximum power output of 8.2 nW at ΔT = 10.8 K. Accordingly, the estimated power density of 0.10 mW cm$^{-2}$ at ΔT = 10.8 K or 2.4 mW cm$^{-2}$ at ΔT = 50 K is comparable with those for $Bi_2Te_3$/$Sb_2Te_3$-based devices[31,32]. The details for simulation of the maximum power output at a certain temperature difference are described in Supplementary Note 3 and shown in Supplementary Fig. 4. We also evaluated the energy conversion efficiency of this module, as described in Supplementary Note 4, and the results are shown in Supplementary Fig. 5. A conversion efficiency of 0.8% for the CuI-based module is obtained at ΔT = 50 K, which is close to that for p-type PbTe-based module (~1%) at the same temperature difference[28].

## Discussion

The above results indicate that the CuI-based thermoelectric module is promising for low-power applications, such as body-heat-powering for wearable devices, thermoelectric windows, and on-chip cooling or power recovery for miniaturized chips. For example, the CuI-based thermoelectric module can generate a power density of $20\,\mu W\,cm^{-2}$ at $\Delta T = 5\,K$, which is sufficient to run wearable devices such as electric watches, accelerometers, ozone sensors, blood-pressure monitors and electrocardiogram sensors. The power consumption of such small devices is approximately $10–150\,\mu W$ (ref. 33), which can be provided by CuI-based thermal batteries with an area of several $cm^2$.

Another example is large-area thermoelectric windows for sustainable renewable power generation. The transparent CuI-based thermoelectric elements could generate energy density of around $0.1\,kWh\,m^{-2}$ every day with a temperature gradient of $20\,K$ between indoor and outdoor environments of a building. Hence, in theory, such a thermoelectric window with an area of about $10\,m^2$ could run a refrigerator all day long (electricity consumption of about $1\,kWh$ per day), saving about 100 Euro per year for a household in Germany (electricity price of 0.297 Euro per kWh, data from Eurostat 2016). For office buildings with generally high window-to-wall ratios, much more electricity expenses would be saved by applying our designed thermoelectric windows.

We compared the bandgap $E_g$ and room-temperature $ZT$ for typical thermoelectric materials in Fig. 4 (see detailed data in Supplementary Table 1). Among transparent materials ($E_g > 3\,eV$), p-type $\gamma$-CuI thin film exhibits a 1,000 times improvement of the $ZT$ compared to those of any other p-type material such as $CuAlO_2$ (ref. 12), and almost twice compared to those of n-type materials, such as Al-doped ZnO[9] and ITO[10]. Nevertheless, the thermoelectric properties of CuI can be further improved: for example, it is possible to enhance the $ZT$ of the CuI by increasing its hole density, which is limited to around $1\times10^{20}\,cm^{-3}$ in this work due to unintentional doping. Taking into account the non-parabolicity of the bands[20] for heavily doped $\gamma$-CuI, a much higher hole density (well above $1\times10^{20}\,cm^{-3}$) is required to improve the power factor and thereby the $ZT$ value. For this purpose, we suggest to use the extrinsic doping method which is widely used to produce highly degenerate semiconductors, for example, degenerating ZnO doped with Al or Ga. Anion dopants such as N, P, O, S and Se can be good candidates for extrinsic doping of the CuI.

Another generic approach for ZT enhancement is to prepare CuI in the form of nanostructures, such as thin-film superlattices[2], quantum-dot superlattices[3], nanocomposites[34] and heterostructures[35]. In our previous work, we have realized nanostructured CuI, as well as the heteroepitaxial growth of CuI thin films on various substrates or semiconductor thin films[16,17]. Consequently, various types of nanostructures and multilayers of CuI could be realized in order to achieve higher ZT values.

In conclusion, transparent p-type $\gamma$-CuI thin films exhibit high thermoelectric performance in room temperature range[36,37]. Large Seebeck coefficients are observed and fitted well with the single-band model. The low-thermal conductivity near room temperature is attributed to the heavy element iodine, as well as the strong phonon scattering within the obtained polycrystalline $\gamma$-CuI thin films. At the same time, these films are transparent and exhibit a high transmission of 60–85% in the visible spectral range. Accordingly, we have achieved a high thermoelectric figure of merit $ZT \sim 0.23$ near room temperature for this transparent film material, which is 1,000 times higher compared to any other p-type transparent material and almost twice compared to n-type materials. A prototype of CuI-based transparent thermoelectric module shows a good mechanical flexibility with a high power density of $2.4\,mW\,cm^{-2}$ at $\Delta T = 50\,K$, which is comparable with those for $Bi_2Te_3/Sb_2Te_3$-based devices. These results show the great potential of $\gamma$-CuI as a transparent flexible thermoelectric material. In combination with available n-type transparent thermoelectric materials, such as Al-doped ZnO and Sn-doped $In_2O_3$, invisible thermoelectric modules can be readily constructed by coupling the transparent n- and p-type thermoelectric legs. Such invisible and flexible thermoelectric element can be used for designs of thermoelectric windows, body-heat-powered wearable devices, and on-chip cooling or power recovery for miniaturized chips. The sputtering technique at room temperature used here can be upscaled for producing large area transparent CuI thin films on low-cost flexible and transparent substrates.

## Methods

**Sample preparation and structural characterization.** Polycrystalline CuI films were deposited on Corning 1737 glass and PET substrates by reactive sputtering at room temperature. A high purity (99.999%) copper disk was used as the sputtering target. Iodine vapour was introduced by a needle valve connected with a heated iodine source. This iodine source was a stainless steel tube filled with iodine particles, which was kept at 180 °C to sustain a sufficient iodine vapour pressure. Argon was introduced to get the total pressure to 0.02 millibar. The experimental details were also described in our previous report[16,17]. The thickness of the CuI thin film was estimated using a Dektak profilometer.

**Property measurements.** Room-temperature Hall effect was measured in Van-der-Pauw geometry with a magnetic field of 0.4 T to determine the carrier type, density and mobility. For investigation of thermoelectric properties, $S$ and $\sigma$ were measured simultaneously under helium atmosphere with a Linseis LSR-3 instrument with NiCr/Ni and Ni contacts and a continuous reverse of the polarity of the thermocouples (bipolar setup). The in-plane thermal conductivities of the CuI thin films were determined using the differential 3ω method[26]. The optical transmission spectra were measured using a PerkinElmer Lambda 40 ultraviolet–VIS-near infrared spectrometer between a wavelength of 200 and 2,000 nm. The thermoelectric power output of the single-leg demonstrator was measured using a hot plate to create a temperature gradient and simultaneously measuring the temperature difference using an infrared camera FLIR E50. The voltage and current generated between two Au metal contacts was measured using a multimeter. The output voltage and power of the module were measured by changing the load resistance and temperature difference near room temperature. The power density was evaluated with a fixed thin film cross-sectional area $(8\times10^{-5}\,cm^2)$.

**Data availability.** The data that support the findings of this study are available from the corresponding author on reasonable request.

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

## Acknowledgements

We thank Holger Hochmuth for technical support with the CuI sputter system, Ulrike Teschner for the optical spectra measurements, and Jörg Lenzner for the surface morphology and chemical composition measurements. This work was funded by Deutsche Forschungsgemeinschaft (GR 1011/28-1) and Universität Leipzig within the program of Open Access Publishing. Richard Y.Q.F. acknowledges the funding support from UK EPSRC EP/P018998/1.

## Author contributions

C.Y. and M.G. conceived and designed the research. C.Y. performed the experiments, analysed the data and wrote the paper. M.K. and M.L. contributed sputtering techniques. D.S. and O.O. conducted the Seebeck effect measurement. H.M.W. contributed the Hall effect measurement. M.B., G.B. and Y.Q.F. carried out the thermal conductivity measurement. All authors discussed the results and commented on the manuscript.

## Additional information

**Competing interests:** The authors declare no competing financial interests.

