## [Peer Review File · Nature Communications]

Reviewers' comments:

Reviewer #1 (Remarks to the Author):

It is a very good article dealing with transparent flexible thermoelectric material based on non-toxic earth-abundant p-type copper iodide thin film.

While, thermoelectric devices that are flexible and optically transparent hold unique promise for future electronics, No actual transparent p-type thermoelectric materials do actually exist.

The current manuscript deals with the development of "invisible" CuI based p-type transparent thermoelectric materials with very impressive optical and thermoelectric properties.

The results are very novel and might be very interesting for the scientific community.

My only minor remark concerns with a broader community which is not familiar with this specific application and class of thermoelectric materials. For this community I would recommend on the Introduction chapter, to introduce a broader sentence in the form of "Several classes of non-transparent thermoelectric materials with attractive thermoelectric properties for renewable power generation applications were recently reported, including tellurides [a-d], half-Heuslers [e,f], and silicides [g,h]" with the following references:

a. Y. Gelbstein, Z. Dashevsky and M.P. Dariel, "In-doped Pb_{0.5}Sn_{0.5}Te p-type Samples Prepared by Powder Metallurgical Processing for Thermoelectric Applications", *Physica B*, 396 16-21 (2007).

b. Yaniv Gelbstein and Joseph Davidow, Highly efficient functional GexPb_{1-x}Te based thermoelectric alloys, *Physical Chemistry Chemical physics* 16(37) 20120-20126 (2014).

c. Yaniv Gelbstein, Phase morphology effects on the thermoelectric properties of Pb_{0.25}Sn_{0.25}Ge_{0.5}Te, *Acta Materialia* 61(5) 1499-1507 (2013).

d. B. Dado, Y. Gelbstein, D. Mogiliansky, V. Ezersky and M.P. Dariel, Structural evolution following spinodal decomposition of the pseudoternary compound (Pb_{0.3}Sn_{0.1}Ge_{0.6})Te, *Journal of Electronic Materials* 39(9) 2165-2171 (2010).

e. K. Kirievsky, M. Shlimovich, D. Fuks and Y. Gelbstein, An ab-initio study of the thermoelectric enhancement potential in nano-grained TiNiSn, *Physical Chemistry Chemical Physics* 16(37) 20023-20029 (2014).

f. K. Kirievsky, Y. Gelbstein, D. Fuks, Phase separation and antisite defects possibilities for enhancement the thermoelectric efficiency in TiNiSn half-Heusler alloys, *Journal of Solid State Chemistry* 203 247-254 (2013).

g. Yatir Sadia, Liron Dinnerman and Yaniv Gelbstein, Mechanical Alloying and Spark Plasma Sintering of Higher Manganese Silicides for Thermoelectric Application, *Journal of Electronic Materials* 42(7) 1926-1931 (2013).

h. Y. Gelbstein, J. Tunbridge, R. Dixon, M.J. Reece, H.P. Ning, R. Gilchrist, R. Summers, I. Agote, M.A. Lagos, K. Simpson, C. Rouaud, P. Feulner, S. Rivera, R. Torrecillas, M. Husband, J. Crossley, and I. Robinson, Physical, mechanical and structural properties of highly efficient nanostructured n- and p- silicides for practical thermoelectric applications, *Journal of Electronic Materials* 43(6) 1703-1711 (2014).

With these minor revisions I recommend on acceptance of this manuscript.

Reviewer #2 (Remarks to the Author):

The authors have reported a series of semiconducting properties of CuI films and focus on thermoelectric performance in this paper

Although ZT values of the CuI thin films are rather high among the p-type transparent semiconductors, the value is far below 1 which is common in other TE materials. TE materials based on transparent oxides are interesting because they are stable even at high temperatures

and enhanced ZT values can be obtained at higher temperatures. However, gamma phase of CuI is not stable at high temperature.

Is there any practical application as TE of CuI, which is possible even ZT=0.2?

Reviewer #3 (Remarks to the Author):

The authors present a research work related to "Transparent Flexible Thermoelectric Material Based on Non-toxic Earth-Abundant p-Type Copper Iodide Thin Film" where the major innovation is the high ZT of transparent thermoelectric at room temperature as the transparency and electrical properties of p-type CuI have been shown in a previous paper of the same authors ref 14 - PNAS 2016 and also by other authors.

Indeed the transparent Thermoelectric materials has been proposed by the authors of reference 6 Loureiro et al. but for n-type AZO transparent conductor. The achievement of a p-type material with similar or higher ZT will open doors towards new field of applications for the transparent thermoelectric materials, therefore important for the transparent electronics community.

The work content is consistent leading to convincing conclusions, the experimental results are supported by theoretical calculations and by other published similar investigations. The complementary information gives details about the CuI films structure and compares results of this work to other materials and authors, therefore highlighting the TE properties of CuI of this work. No remarkable errors or mistakes were seen but the paragraph "CuI undergoes multiple polymorphic phase transitions during heating, i.e., γ (zincblende structure, space group $F\bar{4}3m$) β (wurtzite structure, $P63mc$) α (rock salt structure, $Fm\bar{3}m$) at 643 K and 673 K, respectively" needs a reference.

Response to referees

Reviewer #1:

It is a very good article dealing with transparent flexible thermoelectric material based on non-toxic earth-abundant p-type copper iodide thin film.

While, thermoelectric devices that are flexible and optically transparent hold unique promise for future electronics, no actual transparent p-type thermoelectric materials do actually exist.

The current manuscript deals with the development of "invisible" CuI based p-type transparent thermoelectric materials with very impressive optical and thermoelectric properties.

The results are very novel and might be very interesting for the scientific community.

My only minor remark concerns with a broader community which is not familiar with this specific application and class of thermoelectric materials. For this community I would recommend on the Introduction chapter, to introduce a broader sentence in the form of "Several classes of non-transparent thermoelectric materials with attractive thermoelectric properties for renewable power generation applications were recently reported, including tellurides [a-d], half-Heuslers [e,f], and silicides [g,h]" with the following references:

a. Y. Gelbstein, Z. Dashevsky and M.P. Dariel, "In-doped $\text{Pb}_{0.5}\text{Sn}_{0.5}\text{Te}$ p-type Samples Prepared by Powder Metallurgical Processing for Thermoelectric Applications", *Physica B*, 396 16-21 (2007).

b. Yaniv Gelbstein and Joseph Davidow, Highly efficient functional $\text{GexPb}_{1-x}\text{Te}$ based thermoelectric alloys, *Physical Chemistry Chemical physics* 16(37) 20120-20126 (2014).

c. Yaniv Gelbstein, Phase morphology effects on the thermoelectric properties of $\text{Pb}_{0.25}\text{Sn}_{0.25}\text{Ge}_{0.5}\text{Te}$, *Acta Materialia* 61(5) 1499-1507 (2013).

d. B. Dado, Y. Gelbstein, D. Mogiliansky, V. Ezersky and M.P. Dariel, Structural evolution following spinodal decomposition of the pseudoternary compound ($\text{Pb}_{0.3}\text{Sn}_{0.1}\text{Ge}_{0.6}\text{Te}$), *Journal of Electronic Materials* 39(9) 2165-2171 (2010).

e. K. Kirievsky, M. Shlimovich, D. Fuks and Y. Gelbstein, An ab-initio study of the thermoelectric enhancement potential in nano-grained TiNiSn , *Physical Chemistry*

Chemical Physics 16(37) 20023-20029 (2014).

f. K. Kirievsky, Y. Gelbstein, D. Fuks, Phase separation and antisite defects possibilities for enhancement the thermoelectric efficiency in TiNiSn half-Heusler alloys, Journal of Solid State Chemistry 203 247-254 (2013).

g. Yatir Sadia, Liron Dinnerman and Yaniv Gelbstein, Mechanical Alloying and Spark Plasma Sintering of Higher Manganese Silicides for Thermoelectric Application, Journal of Electronic Materials 42(7) 1926-1931 (2013).

h. Y. Gelbstein, J. Tunbridge, R. Dixon, M.J. Reece, H.P. Ning, R. Gilchrist, R. Summers, I. Agote, M.A. Lagos, K. Simpson, C. Rouaud, P. Feulner, S. Rivera, R. Torrecillas, M. Husband, J. Crossley, and I. Robinson, Physical, mechanical and structural properties of highly efficient nanostructured n- and p- silicides for practical thermoelectric applications, Journal of Electronic Materials 43(6) 1703-1711 (2014).

With these minor revisions I recommend on acceptance of this manuscript.

Reply to Comment:

According to this very favorable comment, we have added the suggested sentence, and also added the references to the introduction. Pls refer to line 7 on page 2, and References lists in Page 13.

=====

Reviewer #2:

The authors have reported a series of semiconducting properties of CuI films and focus on thermoelectric performance in this paper.

Although ZT values of the CuI thin films are rather high among the p-type transparent semiconductors, the value is far below 1 which is common in other TE materials. TE materials based on transparent oxides are interesting because they are stable even at high temperatures and enhanced ZT values can be obtained at higher temperatures. However, gamma phase of CuI is not stable at high temperature.

Is there any practical application as TE of CuI, which is possible even $ZT=0.2$?

Reply to Comment:

The comments are generally positive and in detail very helpful and constructive. Here

are our replies and list of revisions.

(1) Research field and present research stage

In this study, we explored the potential of CuI for thermoelectric energy conversion, and subsequently demonstrated, for the first time, a promising p-type thermoelectric material which is transparent, flexible, non-toxic and earth-abundant. Our finding will surely trigger general interest and further studies will follow in this newly developed field of “transparent thermoelectrics”.

However, it is true that the research on transparent thermoelectric materials, such as p-type CuI, is still at its beginning stage. Much room remains for further improvement of the thermoelectric properties of CuI using various methods and technologies. Every single process step needs to be further optimized in order to improve the ZT value, not only for p-type CuI but also for many other types of n-type transparent oxides. On the other hand, new transparent thermoelectric materials should be explored for different applications. For example, new p-type transparent materials with high thermal stability for high-temperature thermoelectric applications.

(2) Compare ZT of CuI and transparent oxides

In this study, a high room-temperature ZT value of 0.21 for the CuI has been achieved, which is 1000 times greater than that of any other previously tested p-type transparent material, and even higher than those of its n-type counterparts such as $ZT = 0.1$ for Al-doped ZnO (AZO) and $ZT = 0.14$ for In-doped SnO₂ (ITO). Therefore, for thermoelectric applications near room temperature, CuI is by far one of the best transparent materials.

We agree with the reviewer that one of the critical requirements for the transparent oxides is their high thermal stability, thus leading to enhanced ZT values at a high temperature. For example, ZT of AZO increases from 0.1 at 300 K to 0.3 at 1300 K [J. Appl. Phys. **79**, 1816 (1996)]. However, it should be pointed out that the energy conversion efficiency of a thermoelectric device should be significantly dependent on the average ZT values over a wide-range temperature rather than on their maximum value. For the CuI in this study, the averaged ZT value of ~ 0.22 is comparable with that for AZO even at high temperatures.

Nevertheless, we agree that for further exploration of thermoelectric materials, effort should be made to improve the thermoelectric properties of both n- and p-type transparent materials, at room and also at elevated temperatures.

(3) Approaches to enhance ZT of CuI

For CuI, it is not easy to enhance ZT immediately by simply increasing the operation temperature due to its thermal stability. We have done further experimental work to improve the thermal stability of CuI using capping layers. For example, a CuO_x capping layer could significantly suppress the degrading of the conductivity of CuI during air-annealing. These new experimental data have been added in the Supporting Information as Figure S3. However, more efforts should be made in further investigations to improve the qualities of the capping layers by optimizing the deposition methods, and also to explore more suitable capping materials.

On the other hand, it is possible to further improve the thermoelectric properties of the CuI by increasing its hole density. For this purpose, we suggest to use the extrinsic anion-doping method in the future exploration. The hole density of CuI in this work is limited to $\sim 1 \times 10^{20} \text{ cm}^{-3}$ due to unintentional doping (mainly attributed to copper vacancies). However, in theory a much higher hole density ($> 1 \times 10^{21} \text{ cm}^{-3}$) is required to improve the power factor and thereby the ZT value [20]. Therefore, we suggest that extrinsic doping method should be explored and this method is widely used to produce highly degenerate semiconductors, e.g. degenerate ZnO doped with Al or Ga. Anion dopants such as N, P, O, S and Se can be good candidates for extrinsic doping of CuI, while the acceptor levels need to be considered. Although our available facilities are not suitable for controllable anion-doping of CuI, we could foresee the success in the near future by developing appropriate sample preparation methods.

Furthermore, it is possible to enhance ZT values of CuI by designing and realizing its nanostructures, such as thin-film superlattices [2], quantum-dot superlattices [3], nanocomposites [34] and heterostructures [35]. In our previous work, we have realized nanostructured CuI as well as the heteroepitaxial growth of CuI thin films on various substrates or semiconductor thin films [16,17]. Based on these studies, diverse nanostructures and multilayers of CuI could be realized.

(4) Operation temperatures for transparent thermoelectrics

Transparent thermoelectric modules could be served as novel “transparent” power sources to run transparent electronic devices and also many other optoelectronic devices. Most integrated circuits, including those in transparent electronic devices, commonly operate at or near room temperature, and can hardly withstand temperatures over 100 °C. Therefore, transparent thermoelectric modules are commonly operated near room temperature.

In addition, not limited to applications into integrated circuits, there are actually

numerous heat sources such as home heating and human/animal bodies, which are not far away from room temperatures. Hence, thermoelectric materials have found and will find extensive applications in these fields.

From these points of view, in this study, we have focused mainly on the room-temperature thermoelectric properties of CuI for transparent thermoelectric applications.

(5) Some practical applications for materials with $ZT < 1$

Both p-type CuI and n-type transparent oxides show $ZT < 1$, therefore potential applications could be **low-power electronics, such as body-heat-powering for wearable/portable devices, thermoelectric windows, and on-chip cooling or power recovery for miniaturized chips.**

For example, the CuI-based thermoelectric module could generate power density of $20 \mu\text{W}/\text{cm}^2$ with a temperature difference of $5 \text{ }^\circ\text{C}$, which is sufficient to run wearable devices such as electric watches, accelerometers, ozone sensors, blood-pressure monitors and electrocardiogram sensors. The power consumption of such small devices are approximately $10\text{-}150 \mu\text{W}$ [33], which can be provided by CuI-based thermal batteries with area of several cm^2 .

Another example is large-area thermoelectric windows for sustainable renewable power generation. The transparent CuI-based thermoelectric elements could generate energy density of $\sim 0.1 \text{ kWh}/\text{m}^2$ every day with a temperature gradient of 20 K between indoor and outdoor environments of a building. Hence, in theory, such thermoelectric windows with area of about 10 m^2 could run a refrigerator all day long (electricity consumption of $\sim 1 \text{ kWh}/\text{day}$), saving $\sim 100 \text{ Euro}/\text{year}$ for a household in Germany (electricity price of $0.297 \text{ Euro}/\text{kWh}$, data from Eurostat 2016). For office buildings with generally high window-to-wall ratios, much more electricity expenses would be saved by thermoelectric windows.

We have modified the manuscript based on the comments from reviewer 2 and our consideration above. These include:

Line 2 on page 8:

We deleted the sentence: “This power density is applicable for self-powered mobile electronic devices operated near room temperature.”

Line 12 on page 8:

We added the sentence: “For example, as shown in the Supporting Information **Figure S3**, the degradation of the conductivity of γ -CuI thin film during air-annealing could be significantly suppressed by capping with a CuO_x layer. It shows a three times reduced effect of increasing resistance and thus proves that further technological work targeted towards capping can extend the range of useful operation temperatures. Detailed study on the thermal stability of CuI are subject to further investigations by design and development of suitable capping layers.”

Accordingly, in the Supporting Information, we added new paragraph and Figure S3:

“Improved thermal stability of obtained CuI thin Films

In order to improve the thermal stability of γ -CuI thin film, a 30-nm thick CuO_x layer was employed as a capping layer for protecting the CuI sample surface. Such an oxide capping layer could act as a barrier to prevent the iodine diffusion during thermal annealing. Here, the CuO_x layer was deposited by DC sputtering at room temperature using a high-purity (99.999%) copper target. The total amount of gas mixture was fixed to 100 sccm, and the argon-to-oxygen ratio was 5:95. The as-deposited CuO_x layer was amorphous and insulating. Subsequently, the CuI thin films with and without a CuO_x capping layer were annealed in air at various temperatures for 5 min. The sheet resistance R_{sheet} of these samples are shown in **Figure S3**. It shows a three times reduced effect of increasing resistance and thus proves that further technological work targeted towards capping can extend the range of useful operation temperatures.”

Line 10 on page 10:

“These results indicate that the CuI-based thermoelectric module is promising for applications at high heat flows and small temperature differences, such as the biothermal battery for cardiac pacemaker.”

The above sentence has been modified thoroughly with the following paragraph:

→

“These results indicate that the CuI-based thermoelectric module is promising for low-power applications, such as body-heat-powering for wearable devices, thermoelectric windows, and on-chip cooling or power recovery for miniaturized chips. For example, the CuI-based thermoelectric module can generate a power density of $20 \mu\text{Wcm}^{-2}$ at $\Delta T = 5 \text{ K}$, which is sufficient to run wearable devices such as electric watches, accelerometers, ozone sensors, blood-pressure monitors and electrocardiogram sensors. The power consumption of such small devices is approximately $10\text{-}150 \mu\text{W}^{33}$, which can be provided by CuI-based thermal batteries with an area of several cm^2 .”

Another example is large-area thermoelectric windows for sustainable renewable power generation. The transparent CuI-based thermoelectric elements could generate energy density of $\sim 0.1 \text{ kWhm}^{-2}$ every day with a temperature gradient of 20 K between indoor and outdoor environments of a building. Hence, in theory, such a thermoelectric window with an area of about 10 m^2 could run a refrigerator all day long (electricity consumption of $\sim 1 \text{ kWh/day}$), saving $\sim 100 \text{ Euro/year}$ for a household in Germany (electricity price of 0.297 Euro/kWh , data from Eurostat 2016). For office buildings with generally high window-to-wall ratios, much more electricity expenses would be saved by applying our designed thermoelectric windows.”

Line 7 on page 11:

“It is possible to further enhance the thermoelectric performance of the γ -CuI thin films by optimizing the film mobility and carrier density, or nanostructuring to reduce phonon thermal conductivity.”

→

“Nevertheless, the thermoelectric properties of CuI can be further improved: for example, it is possible to enhance the ZT of the CuI by increasing its hole density, which is limited to $\sim 1 \times 10^{20} \text{ cm}^{-3}$ in this work due to unintentional doping. Taking into account the non-parabolicity of the bands²⁰ for heavily-doped γ -CuI, a much higher hole density ($\gg 1 \times 10^{20} \text{ cm}^{-3}$) is required to improve the power factor and thereby the ZT value. For this purpose, we suggest to use the extrinsic doping method which is widely used to produce highly degenerate semiconductors, e.g. degenerating ZnO doped with Al or Ga. Anion dopants such as N, P, O, S and Se can be good candidates for extrinsic doping of the CuI.

Another generic approach for ZT enhancement is to prepare CuI in the form of nanostructures, such as thin-film superlattices², quantum-dot superlattices³, nanocomposites³⁴ and heterostructures³⁵. In our previous work, we have realized nanostructured CuI as well as the heteroepitaxial growth of CuI thin films on various substrates or semiconductor thin films^{16,17}. Consequently, various types of nanostructures and multilayers of CuI could be realized in order to achieve higher ZT values.”

Line 12 on page 12:

“Such ‘invisible’ and flexible thermoelectric element can be used for design of a smart window, or integration with diverse optoelectronic devices such as flexible solar cells

and displays for fast on-chip cooling and power recovery.”

→

“Such ‘invisible’ and flexible thermoelectric element can be used for design of thermoelectric windows, body-heat-powered wearable devices, and on-chip cooling or power recovery for miniaturized chips.”

The following references have been added:

[33] Hyland, M., Hunter, H., Liu, J., Veety, E., & Vashaee, D. Wearable thermoelectric generators for human body heat harvesting. *Appl. Energy* **182**, 518–524 (2016).

[34] Biswas, K. *et al.* High-performance bulk thermoelectrics with all-scale hierarchical architectures. *Nature* **489**, 414–418 (2012).

[35] Shakouri, A. & Bowers, J. E. Heterostructure integrated thermionic coolers. *Appl. Phys. Lett.* **71**, 1234-1236 (1997).

Reviewer #3:

The authors present a research work related to "Transparent Flexible Thermoelectric Material Based on Non-toxic Earth-Abundant p-Type Copper Iodide Thin Film" where the major innovation is the high ZT of transparent thermoelectric at room temperature as the transparency and electrical properties of p-type CuI have been shown in a previous paper of the same authors ref 14 - PNAS 2016 and also by other authors.

Indeed, the transparent Thermoelectric materials has been proposed by the authors of reference 6 Loureiro et al. but for n-type AZO transparent conductor. The achievement of a p-type material with similar or higher ZT will open doors towards new field of applications for the transparent thermoelectric materials, therefore important for the transparent electronics community.

The work content is consistent leading to convincing conclusions, the experimental results are supported by theoretical calculations and by other published similar investigations. The complementary information gives details about the CuI films

structure and compares results of this work to other materials and authors, therefore highlighting the TE properties of CuI of this work.

No remarkable errors or mistakes were seen but the paragraph "CuI undergoes multiple polymorphic phase transitions during heating, i.e., γ (zincblende structure, space group $F\bar{4}3m$) β (wurtzite structure, $P63mc$) α (rock salt structure, $Fm\bar{3}m$) at 643 K and 673 K, respectively" needs a reference.

Reply to Comment:

We welcome the appreciation of our work.

According to the comment, we add the references for the above mentioned paragraph:

[18] Keen, D. A. & Hull, S. The high-temperature structural behaviour of copper(I) iodide. *J. Phys.: Condens. Matter* **7**, 5793-5804 (1995).

Reviewers' comments:

Reviewer #2 (Remarks to the Author):

Reviewer #3 (Remarks to the Author):

The paper is OK.